# A Glimpse on the Evolution of RNA Viruses: Implications and Lessons from SARS-CoV-2

**DOI:** 10.3390/v15010001

**Published:** 2022-12-20

**Authors:** Petra Šimičić, Snježana Židovec-Lepej

**Affiliations:** Department of Immunological and Molecular Diagnostics, University Hospital for Infectious Diseases “Dr. Fran Mihaljević”, HR-10000 Zagreb, Croatia

**Keywords:** RNA virus, evolution, quasispecies, mutation, recombination, coronavirus, SARS-CoV-2

## Abstract

RNA viruses are characterised by extremely high genetic variability due to fast replication, large population size, low fidelity, and (usually) a lack of proofreading mechanisms of RNA polymerases leading to high mutation rates. Furthermore, viral recombination and reassortment may act as a significant evolutionary force among viruses contributing to greater genetic diversity than obtainable by mutation alone. The above-mentioned properties allow for the rapid evolution of RNA viruses, which may result in difficulties in viral eradication, changes in virulence and pathogenicity, and lead to events such as cross-species transmissions, which are matters of great interest in the light of current severe acute respiratory syndrome coronavirus 2 (SARS-CoV-2) pandemics. In this review, we aim to explore the molecular mechanisms of the variability of viral RNA genomes, emphasising the evolutionary trajectory of SARS-CoV-2 and its variants. Furthermore, the causes and consequences of coronavirus variation are explored, along with theories on the origin of human coronaviruses and features of emergent RNA viruses in general. Finally, we summarise the current knowledge on the circulating variants of concern and highlight the many unknowns regarding SARS-CoV-2 pathogenesis.

## 1. Introduction

RNA viruses are obligate intracellular parasites characterised by extremely high genetic variability and phenotypic diversity, facilitating infection of an extensive range of hosts [1,2,3]. Significantly different genome structures and replication strategies allow for great adaptability and exploitation of various host cellular mechanisms [4,5,6,7]. One of the prerequisites of successful adaptation to various hosts and environments is the ability to efficiently introduce genetic change in a short amount of time [8,9]. Alterations in the virus’s genetic material may result in changes in the virus phenotype and population dynamics, leading to events such as a change in virulence or host tropism, potentially resulting in new emergent pathogens [10,11,12].

The outbreak of severe acute respiratory syndrome coronavirus 2 (SARS-CoV-2) in 2019 and the ongoing pandemics once again emphasised the importance of elucidating molecular mechanisms of RNA genome evolution and potential viral emergence [13,14,15]. Coronaviruses, which constitute the subfamily *Orthocoronaviridae* among the *Coronaviridae* family, were shown to be distributed among various animals and capable of causing diseases with a broad range of symptoms and degrees of pathology [16,17,18]. However, many of the evolutionary, structural, and functional features are shared and sometimes intertwined among the coronaviruses, enabling the analysis of the genetic structure change in entire populations [12,19,20,21].

The first part of this review article aims to summarise the main RNA virus characteristics necessary for understanding the mechanisms and causal agents that govern RNA virus variation. The second part applies the above-mentioned concepts to the evolution of human coronaviruses with an emphasis on the emergence of SARS-CoV-2. Finally, the unresolved questions and long-term consequences of the rapid RNA virus evolution are discussed as we witness the aftermath of one of the largest globally known pandemics.

## 2. RNA Virus Classification

### 2.1. Baltimore Classification

In 1971 David Baltimore published an article titled *Expression of Animal Virus Genomes*, suggesting a division of viruses into six classes according to the method of transmission of their genetic information from one generation to another (mRNA synthesis) and the style of expression of their genetic information (structure of nucleic acid) [4]. The original Baltimore classification placed viruses into one of six classes—class I (double-stranded DNA genome, (+/−) dsDNA), class II (single-stranded DNA genome, (+) ssDNA), class III (double-stranded RNA genome, (+/−) dsRNA), class IV (positive sense single-stranded RNA genome, (+) ssRNA), class V (negative-sense single-stranded RNA genome, (−) ssRNA), and class VI (reverse-transcribing single-stranded RNA genome, (+) ssRNA-RT). The polarity of the genome is designated as “+” and “−” strands, or strands of both polarities (“+/−”) [4]. This classification was later updated with class VII to accommodate the viruses with a double-stranded DNA genome with an RNA intermediate in its replication cycle ((+/−) dsDNA-RT) [22] (Figure 1).

This simple yet functional and intuitive classification system is still used today since it nicely complements virus taxonomy [18]. Unlike cellular life forms, which are strictly dependent on the replication of the dsDNA genome, the abundance of viral RNA and DNA replication–expression strategies enables their stark genetic and phenotypic diversity [4,23]. True RNA viruses with no DNA intermediate in their replication cycle comprise 3 out of 7 Baltimore classes (III, IV, and V) and could be considered one of the earliest descendants from the primordial genetic pool [24,25]. All three of the above-mentioned Baltimore classes include viruses with non-segmented and segmented genomes. A segmented genome can be packed into single virions resulting in true segmented viruses, or into multiple virions, resulting in multipartite viruses [26,27]. It is important to note that recent advances in evolutionary dynamics analyses confirmed that Baltimore classes do not accurately reflect evolutionary relationships among viruses, discerning the implicit presumption that each class is of monophyletic origin [28].

### 2.2. The International Committee on Taxonomy of Viruses Classification

In 2019 and 2020, the International Committee on Taxonomy of Viruses (ICTV), a global organisation responsible for developing virus taxonomy and nomenclature, introduced a new fifteen-rank classification hierarchy of virus taxonomy to replace the previously used five-rank hierarchy of species, genus, subfamily, family, and order. The 15-rank classification includes eight primary ranks (realm, kingdom, phylum, class, order, family, genus, and species) and seven secondary ranks, which resemble those used in Linnean systems [18]. Realm *Riboviria* was established as a likely monophyletic clade of RNA viruses that use RNA-directed RNA polymerase for genome replication, which includes viruses from 3 out of 7 Baltimore classes (III, IV, and V) [18,29]. The realm *Riboviria* was later extended to include nearly all RNA viruses and reverse-transcribing viruses, with a total of six realms recognised up to this day [30,31]. Even though some of the Baltimore classes do seem to be strictly monophyletic (such as class V, (−) ssRNA viruses) and others are clearly polyphyletic (such as class I, (+/−) dsDNA viruses), some form paraphyletic taxons with respect to other classes and therefore cannot be considered traditionally either (such as class III, dsRNA, with respect to class V, (−) ssRNA) [28]. With time, new discoveries obtained by metaviromics will inevitably change the hierarchical taxonomy of viruses. It is important to note that even though it was shown that Baltimore classes could not be considered taxonomic categories, both classification systems are helpful and complement each other well, especially for deciphering the means of viral evolution.

## 3. RNA Virus Characteristics

### 3.1. Host Range

RNA viruses are primarily infectious agents of Eukarya [32]. No known RNA viruses that infect Archaea were known until the last decade. At the same time, recent findings suggest that RNA viruses detected in archaeal hosts could be direct ancestors of eukaryotic RNA viruses [1]. RNA viruses of bacteria most commonly have a dsRNA genome; however, the taxonomy of bacteria-infecting ssRNA viruses was recently expanded [2]. Nevertheless, the majority of the viruses infecting prokaryotes have DNA genomes, particularly Baltimore class I ((+/−) dsDNA), with RNA viruses accounting for a minority of virome diversity [33]. In contrast to Archaea and Bacteria, eukaryotes are a host for the vast majority of RNA viruses, particularly of Baltimore class IV ((+) ssRNA) [3].

### 3.2. Structural Genome Features

Even though RNA viruses are known for their large structural and functional diversity, their genome sizes exhibit relatively small variations with differences within only one order of magnitude. While the smallest RNA viruses measure around 2 kb in size, the maximum observed genome length of 30–40 kb is reached among the *Coronaviridae* family [34,35]. This is primarily due to physical and structural constraints on RNA genome stability and RNA polymerase fidelity limitations with larger genomes consisting exclusively of class I dsDNA viruses [22,23,34,36,37]. One of the consequences of the small genome size is genome compression, which commonly manifests as gene overlap [38,39]. A universally shared structural characteristic of RNA viruses is the presence of untranslated regions (UTR) surrounding one or more open reading frames (ORF) at both 5′ and 3′ ends, usually containing sequences or conserved structures needed for replication regulation [40].

### 3.3. RNA Regulatory Processes

RNA-dependent RNA polymerase is an essential enzyme for viral RNA replication because it catalyses RNA replication from an RNA template. It is a process generally not typical in eukaryotic cells, except in events such as RNA-mediated silencing pathways and telomere formation [41]. Therefore, RNA viruses from Baltimore classes III and V encode their own RNA-dependent RNA polymerase and incorporate it into virions. The exception is viruses from class IV, which carry a positive single-stranded RNA genome that cellular machinery can directly transcribe upon viral entry [22]. Regulation of RNA replication, a fundamental step in the virus replication cycle, primarily depends on the structure of the viral RNA genome. Viruses with dsRNA genomes must synthesise both mRNA (which can occur as a direct transcription of the viral genome) and viral RNA (which is generated via (−) RNA strand intermediate). Similarly, (+) ssRNA genomes may already act as mRNA but still require (−) ssRNA intermediate for generating viral RNA. On the other hand, viruses with (−) ssRNA genomes, which are complementary in base sequence to the mRNA, first require its transcription to produce the (+) ssRNA, which may act as both mRNA and a template for genome replication. A delicate regulation balance between replication and transcription processes is vital for the successful synthesis of new viral progeny [4,22,42].

Canonical posttranscriptional processing necessary for the maturation of the primary transcript in cellular organisms includes obtaining a 5’-cap, splicing, and generation of a 3’ poly-A tail [43]. By mimicking the mechanisms mentioned above, viruses could increase the probability of perseverance of their genetic information during replication. Some RNA viruses use cap-snatching or ‘stealing’ of the cap from cellular mRNA; some use viral replication machinery to synthesise cap equivalent, while others disregard the use of 5’-cap completely, as reviewed in Decroly et al. (2012) [44]. Alternative polyadenylation signals, such as short poly-U or poly-A stretches, or the presence of highly conserved secondary structures, were observed [45,46,47]. The occurrence of splicing is generally not common in RNA viruses whose genome rarely contains introns [48]. The above-mentioned modifications may be applied to both whole-genome transcripts and nested sets of subgenomic mRNAs, which result from discontinuous transcription characteristic for members of the *Coronaviridae* family [49,50].

While the degree of independence of transcription and posttranscriptional processing in RNA viruses is a result of the intrinsic viral characteristics, translation is ultimately dependent on the host cell translation machinery. No matter if their genome may be used directly as mRNA or if the transcription of the minus strand must occur first, the translation of the resulting viral mRNA is completely dependent on cellular ribosomes. Since the cellular translation machinery is rather complex, viruses exhibit numerous tactics of manipulation, from an attack on regulatory signalling pathways to targeting specific steps of the translation process itself, comprehensively reviewed by Jaafar et al. (2018) and Jan et al. (2016) [51,52]. Furthermore, many viruses developed specific mechanisms in order to bypass the canonical phases of translation, such as the utilisation of the internal ribosome entry sites (IRES), the use of ribosomal frameshifting signals, and leaky scanning, which simultaneously contribute to the increase in the coding capacity of the rather small RNA virus genomes [53,54]. All of the properties and strategies mentioned above result in the successful hijacking of host cell machinery for the optimisation of the production of a large amount of viral progeny in a very short time, which is another hallmark of virus replication [55].

### 3.4. Quasispecies Concept

The concept of quasispecies was first developed by Manfred Eigen and Peter Schuster in the search of an adequate model for the origin and evolution of early life forms, defining quasispecies as a distribution of closely related replicative units centred around one or several (degenerate) master copies [56,57,58]. Before the introduction of the quasispecies concept, the main target of the (genetic) selection was considered to be the ‘wild type’ population. The quasispecies concept emphasises that selection does not apply to a single individual ‘wild type’ genome but rather within the ‘wild type’ distribution of closely related mutant genomes [59]. The error threshold seems to be a consequence of quasispecies existence and simultaneously the constraint of quasispecies evolution [59,60]. Viruses operate closely to their error threshold, which enables the greatest genetic variation; however, a slight change, such as a decrease in replication fidelity, can push the whole viral population beyond the error threshold into extinction [59,60,61]. The concept of quasispecies was further developed in recent years, emphasising real quasispecies in contrast to the theoretical model. The first experimental evidence of quasispecies was observed by analysing nucleotide sequences of bacteriophage Qβ. It was concluded that no unique structure of the ‘wild type’ bacteriophage Qβ genome existed, but rather, a large number of individual mutant sequences in dynamic equilibrium [62]. Furthermore, since the scope of viral pathogenesis could hardly be predicted due to the everchanging and fleeting quasispecies dynamics, possible medical implications of quasispecies existence were proposed [63]. The quasispecies nature of the virus population was soon demonstrated for various RNA virus populations, such as vesicular stomatitis virus and hepatitis C virus [63,64,65]. The scope of quasispecies was later broadened to refer to “dynamic distributions of non-identical but closely related mutant and recombinant viral genomes subjected to a continuous process of genetic variation, competition and selection, and which act as a unit of selection” [66].

## 4. Mechanisms of RNA Virus Variation

The most important mechanisms responsible for genetic change and, therefore, the rapid evolution and quasispecies nature of RNA viruses are mutations, recombination, and gene segment reassortment [67,68,69]. In addition to the enhancement of viral diversity by introducing genetic change on a greater level than by mutation alone, viral recombination and reassortment can also be one of the mechanisms of repair of genomic molecules [70]. However, for both viral recombination and reassortment to occur successfully, several predispositions have to be met: at least two different viral genomes must be present in the same intracellular area, the resulting recombinant or reassortant must be replication-competent and form infectious viral particles, and finally, the recombinant and reassortant virions must have features that favour their selection among the viral population [71,72]. Other less commonly observed genetic change mechanisms among RNA viruses are gene duplication and gene transfers [73].

### 4.1. Mutation

RNA-dependent RNA polymerase was identified as one of the viral hallmark genes responsible for critical functions in virion structure and genome replication, but is missing from the cellular genome [3,24,74,75]. The origin of RNA-dependent polymerase is intrinsically associated with the origin and evolution of life in general because RNA molecules were considered self-sufficient in the primordial RNA world, with elements such as self-splicing introns speeding up the evolution of hypothetical ribozymes [76]. It is important to note that one of the most conserved protein domains in RNA biogenesis in all kingdoms of life—the RNA recognition motif—is related to the structural fold of the catalytic domain of RNA-dependent RNA polymerase [77]. Eukaryotic RNA-dependent RNA polymerases, however, do not share an evolutionary relationship with viral enzymes [78]. Viral RNA polymerases exhibit a relatively low fidelity of 10^−3^ to 10^−5^ per nucleotide polymerised, leading to the introduction of many mutations at every replication cycle due to the absence of 3’ to 5’ exonucleolytic proofreading activity [66]. However, new revelations suggest that early RNA polymerases had a feature of both proofreading and repair due to the need for extensive information contained within the relatively small genome size of the last universal common ancestor (LUCA) [37]. Low replication fidelity is thought to be one of the factors contributing to the RNA virus’s vast diversity. It is worth mentioning that both an increase and decrease in replication fidelity greater than 4-fold have a negative impact on viral phenotype [79].

### 4.2. Recombination

Recombination is a process of exchange of genetic material between different non-segmented genomes, which is one of the main drivers of RNA virus heterogeneity [66]. The role of recombination in RNA viruses was first observed among the *Picornaviridae* family (such as the poliovirus) as an ‘exception’, but it was soon found to be relatively common, especially among Baltimore class IV (+) ssRNA viruses [80,81,82,83]. Recombination may act as a significant evolutionary force among viruses by creating new potential genome combinations [66]. However, the threshold for the selection of recombinants seems relatively high, with the majority of recombinant viruses being pruned away due to unfavourable properties [84]. Two mechanisms of recombination observed among RNA viruses are replicative recombination, which occurs during RNA synthesis via template switching of the viral polymerase according to the widely accepted copy-choice mechanism, and nonreplicative recombination by breaking and rejoining of the RNA molecules [68,82,85].

On the other hand, homologous recombination occurs between two similar RNA molecules with substantial sequence homology at the same or comparable sites on the paternal strains, while nonhomologous recombination is used to refer to genetic crossover between RNA molecules with no sequence homology [68]. It should be emphasised that even though replicative and nonreplicative recombination mechanisms can theoretically result in both homologous and nonhomologous recombinants, which are ultimately indistinguishable as end products, the vast majority of homologous RNA recombination occurs due to replicative mechanisms because sequence similarity acts as a guideline for the copy-choice mechanism [71,72,85,86]. The fidelity of RNA-dependent RNA polymerase was shown to be a determining factor inversely correlated with replicative recombination frequency [87,88]. Secondary structures of the RNA template, such as hairpins and loops, were also linked to the frequency of replicative recombination; however, data regarding various viruses are conflicting, and no universal control mechanism was established [89,90,91]. The frequency of recombination varies greatly among RNA viruses, with no apparent relation to genome type or replication strategy [83,92].

### 4.3. Reassortment

Another form of genetic exchange sometimes referred to as pseudo-recombination, a feature of all segmented RNA viruses, is gene segment reassortment. Reassortment is a consequence of the co-infection of a single host with two or more segmented viruses, which may result in progeny that contains novel genome combinations derived from both parental genomes [69,93]. Early reassortment models favour random packaging of segments into virions, with the probability of the generation of viable reassortants being determined entirely by chance [94]. Further research suggested that, while this stochastic model probably accurately describes reassortment among multipartite viruses who pack their segmented genome into multiple virus particles, genome segment exchange among viruses who pack their segmented genome into single virions is most likely guided by specific packaging signals [26,95,96]. Unlike recombination, which can in theory occur on any given genome segment and lead to the production of deleterious or nonfunctional proteins, the exchange of entire segments during reassortment ensures the maintenance of functional gene products [26,68]. The frequency of reassortment varies among segmented viruses—while it is relatively common in some viruses, such as influenza A, in other viruses, such as hantaviruses, it seems to occur less frequently [93,97]. It is worth mentioning that non-multipartite segmented viruses usually have meagre recombination rates, making reassortment particularly important for exploring available sequence space [98,99].

## 5. Causes and Consequences of RNA Virus Variation

High mutation rates observed in RNA viruses can be considered pillars of fast RNA virus evolution [66]. However, it should be emphasised that high mutation rates may simultaneously hinder virus emergence and adaptability by not allowing genotypes with potentially advantageous mutations to linger long enough to become fixed in a viral population due to strong selection against deleterious changes [100,101]. Therefore, the notion that a high mutation rate unequivocally enables faster viral adaptation was questioned, and other causes for high mutation rates were explored, such as selection for the robustness of the viral population and selection for fast replication [102]. On the other hand, several theories were proposed to explain the evolutionary reasons for recombination occurrence in RNA viruses. Even though recombination may serve a role in the repair of defective genomes, it does not seem likely that recombination evolved purely for purging detrimental mutations [103]. One theory suggests that recombination is a consequence of genome organisation and the means of RNA replication. Rarely observed recombination events in (−) ssRNA viruses could be attributed to the existence of the ribonucleoprotein complex as a unit of replication, lowering the probability of template switching required for recombination [104,105]. On the other hand, some transcription strategies observed among (+) ssRNA viruses, such as subgenomic RNA transcription, may be responsible for high recombination rates due to the more frequent occurrence of template switching events [79]. However, significant differences in recombination frequency were observed in many viruses with similar genome organisation, such as members of the *Flaviviridae* family [86,106,107]. Nevertheless, the ability to recombine provides many RNA viruses with rapid phenotype change, which can potentially enable further alterations in host tropism, evasion of host immune response, development of resistance to antiviral drugs, or changes in virulence and pathogenicity potentially leading to cross-species transmission [7,12,108].

## 6. Evolution of Human Coronaviruses

### 6.1. Classification of Coronaviruses

Coronaviruses are a group of viruses with the (+) ssRNA genome of Baltimore class IV under the realm of *Riboviria*, which belong to the order *Nidovirales,* family *Coronaviridae,* and subfamily *Orthocoronavirine* [18]. Phylogenetic analyses showed the division of the *Orthocoronavirinae* subfamily into four major genera (*Alphacoronavirus,* Alpha-CoV; *Betacoronavirus*, Beta-CoV, *Gammacoronavirus*, Gamma-CoV, and *Deltacoronavirus,* DeltaCoV) and 26 subgenera, with 52 currently recognised species [109]. Coronaviruses infect a wide variety of vertebrates, with bats and birds being notable reservoirs for Alpha-CoV to Beta-CoV and Gamma-CoV to Delta-CoV, respectively [16]. Even though coronaviruses in animals may cause a wide variety of symptoms, most infections typically result in respiratory or gastrointestinal illnesses [17]. However, due to the sometimes severe and lethal consequences, coronaviruses were more of interest in veterinary medicine, whereas recent epidemics in the 21st century shifted the focus to human coronaviruses (HCoV) [110]. There are currently seven known species or strains of coronaviruses which infect humans, all from Alpha-CoV or Beta-CoV genera, including *Human coronavirus 229E* (HCoV-229E), human coronavirus OC43 (HCoV-OC43), *Human coronavirus NL63* (HCoV-NL63), *Human coronavirus HKU1* (HCoV-HKU1), *Middle East respiratory syndrome-related coronavirus* (MERS-CoV), severe acute respiratory syndrome coronavirus (SARS-CoV) and, as of most recently, severe acute respiratory syndrome coronavirus 2 (SARS-CoV-2) [32,109]. Major epidemics of novel human coronaviruses in the last two decades resulted in multiple changes in *Coronaviridae* taxonomy to accommodate emergent viruses (Figure 2).

The first identified human coronaviruses were HCoV-229E from the genus Alpha-CoV and HCoV-OC43 from the genus Beta-CoV isolated during the 1960s, which were later found to be distributed globally and manifest primarily with respiratory symptoms, such as a common cold [111,112]. An outbreak of severe acute respiratory syndrome (SARS) was reported in 2002–2003, originating in Guangdong province in China and later spreading to other Asian countries, Europe, and North America, with the causative agent identified to be SARS-CoV having more than 8000 confirmed cases and mortality of around 10% [113,114,115]. Subsequent research on coronaviruses led to the discovery of HCoV-NL63 from the genus Alpha-CoV in 2004 and HCoV-HKU1 Beta-CoV in 2005 in patients with respiratory illnesses [116,117]. Both viruses are distributed globally and, together with HCoV-229E and HCoV-OC43, are responsible for 10–29% of common colds [118]. Another ongoing coronavirus outbreak started in 2012 in the Arabian Peninsula and was found to be caused by MERS-CoV, with more than 2600 cases reported as of November 2022 and high mortality exceeding 30% [119,120,121]. However, none of the previous outbreaks were even remotely comparable in scale, as was the third epidemic introduction of coronavirus in the human population, which started in late 2019 in Wuhan, China, and was later attributed to the novel SARS-CoV-2 [14,16]. SARS-CoV-2 is now the seventh coronavirus known to infect humans and was designated as a sister strain of SARS-CoVs of the species *Severe acute respiratory syndrome-related coronavirus* (Figure 2) [122]. As of November 2022, more than 630,000,000 confirmed cases of SARS-CoV-2, including more than 6,600,000 deaths, were observed with enormous socioeconomic consequences [123].

### 6.2. Structure and Genome of Coronaviruses

Coronaviruses have one of the largest genomes among RNA viruses, ranging from 26 to 32 kb, which is almost double the size of the other RNA viruses with large genomes [20,124]. The highly conserved genomic organisation is characteristic of all coronaviruses whose (+) ssRNA genomes end with 5’-cap and 3’ poly-A tail and consist of multiple ORFs often preceded by transcriptional regulatory sequences (TRS). ORFs are surrounded by terminal untranslated regions (UTR) rich in secondary structures with critical regulatory functions necessary for viral replication and transcription [19]. The first two-thirds of the coronavirus genome consist of two major ORFs, ORF1a, and ORF1b, and encode for large replicase polyproteins 1a (PP1a) and C-terminally extended 1b (PP1ab), which are processed into 16 nonstructural proteins (nsp) crucial for coordinating various intercellular aspects of coronavirus replication, including cleavage of polyproteins, vesicle membrane formation, excision of misincorporated nucleotides, mRNA processing, and modulation of host responses [125,126]. The final third of the coronavirus genome encodes for (depending on the coronavirus species) at least four structural proteins (spike, envelope, membrane, and nucleocapsid), which are a part of the infectious viral particle, interspersed with a various number of ORFs encoding for accessory proteins [19,125]. Translation of nonstructural proteins is enabled by ribosome frameshifting, while the expression of structural and accessory proteins is initiated after the generation of 3’ nested subgenomic mRNAs, which all contain a typical leader sequence on the 5’ end by discontinuous transcription guided by TRS [49,50].

### 6.3. Mutation in Coronaviruses

Replication fidelity is one of the main constraints on RNA virus genome size since large genomes generally introduce more mutations in each replication cycle, which threatens to push the virus over the error threshold [59,60]. One of the unique features of coronaviruses is the presence of 3’–5’ exonuclease (ExoN) in nsp14, which was shown to be indispensable for their RNA synthesis [21,126,127,128]. It was suggested that ExoN function was acquired by an ancestral virus that already had a means of successfully replicating the above-average size genome and was critical for its maintenance and eventual enlargement [115,124]. Engineering of CoV nsp14-ExoN viable mutants by substitutions at the ExoN motifs I, II, and III of the nsp14 demonstrated more than a 20× increase in mutation frequency during replication in vitro and a drastic reduction in the accumulation of viral RNA along with defects in sgRNA synthesis [126,127,129]. Exon I motifs (DE), II (D), and III (D) are conserved in the DEDD family of 3’–5’ exonuclease, named after four invariant acidic residues present in cellular organisms that catalyse DNA proofreading [130]. The contribution of nsp14-ExoN to the increase in coronavirus replication fidelity and faithful genome replication is undeniable; however, precise mechanisms still need to be completely elucidated. Several models were proposed, such as direct 3’–5’ proofreading of the nsp14-exoN analogous to cellular DNA mechanisms, direct or indirect stimulation of the intrinsic 3’–5’ activity of RdRp, or potential regulation of RNA recombination [79]. Several studies showed a significant variation in the missense mutation rate along the SARS-CoV-2 genome, with the minimum mutation rate observed in essential regions of SARS-CoV-2 proteins, such as RNA replication machinery and with higher rates of mutation corresponding to structurally more relaxed regions such as spike protein, which was found to be a target of both purifying and positive selection [131,132]. Mutations in the receptor binding domain of spike proteins may lead to the emergence of new variants with changes in fitness, transmission efficacy, or influence on host response and virus neutralisation [133]. The mutation rate for SARS-CoV-2 was found to be between 1 and 5 × 10^−6^ per nucleotide polymerised or of the order of 0.1 per genome per infection cycle [134].

### 6.4. Recombination in Coronaviruses

One of the first occurrences of RNA recombination in viruses with non-segmented RNA genomes was observed on murine coronaviruses. A high frequency of recombination (up to 25%) among RNA genomes of different strains of coronaviruses was shown during mixed infection of susceptible cells, with some of the recombinants appearing to go through multiple crossover events [68,135,136]. Even though mechanisms of coronavirus recombination were not fully understood at the time, it was evident that RNA recombination must have an essential role in virus evolution. It was shown that some recombinants could have evolutionary advantages allowing them to survive or even become dominant in a mixed virus population [68,136]. Early studies also showed that some coronavirus genome regions could be more prone to recombination than others, the so-called recombination ‘hot-spots’ [89]. The large genome size of coronaviruses prone to accumulation of deleterious mutations was considered one of the main reasons for the observed RNA recombination frequency as a means of genome diversification and repair [68,137].

On the other hand, the discontinuous transcription and generation of subgenomic mRNAs by RNA-dependent RNA polymerase used by coronaviruses as a strategy of gene expression resembles the mechanism of copy-choice RNA recombination via template switching and creates a predisposition for such events to occur [49,72]. Therefore, the rate of recombination observed among coronaviruses was found to be the highest among non-segmented (+) ssRNA viruses, with the frequency of mutants being 25% or more during mixed infection with different strains [136,138]. Such high recombination frequency could be partially attributed to the mutational constraint imposed by the proofreading capacity of nsp14-ExoN and the consequentially lower mutation rate, making recombination one of the main mechanisms of exploring available sequence space [139]. Spike glycoprotein gene, one of the main determinants of coronavirus host range and principal factors in viral entry, was identified as a ‘hot-spot’ for recombination among Alpha-CoV and Beta-CoV, allowing for the robust interchange of protein-coding sequences without the loss of infectivity [12,139]. Such changes in spike protein may alter spike–receptor interaction and influence host range, leading to cross-species transmission. The highly recombinogenic nature of coronaviruses from different hosts seems to be one of the critical factors in the origin of SARS-CoV-2, the mechanisms of which are further described in the following chapter [140].

### 6.5. Origin of Human Coronaviruses

With current evidence supporting the hypothesis that bat coronaviruses are gene sources for Alpha-CoV and Beta-CoV, while bird coronaviruses are gene sources for Gamma-CoV and Delta-CoV, recent studies showed that the most recent common ancestor of all coronaviruses dates to around 8100 BC, with no concluding proof of whether it first occurred in bats and later jumped to birds or vice versa [16,141]. All seven coronaviruses currently present in humans emerged from animal reservoirs; however, it is clear that some of the common human coronaviruses, such as HCoV-229E, HCoV-OC43, HCoV-NL63, and HCoV-HKU1 adapted to their new host quite successfully [118,142]. Screening of bats showed the presence of closely related sequences to HCoV-NL63 and HCoV-229E, which suggests they have a bat origin [143,144]. Furthermore, zoonotic recombination of African bat NL63-like viruses and 229E-like viruses in spike protein genes most likely resulted in the ancestor of the HCoV-NL63, while it seems the ancestor of HCoV-229E infected camelids as intermediate hosts [144,145]. Phylogenetic analyses showed the role of rodents in the emergence of HCoV-OC43 and HCov-HKU1 [146]. On the other hand, SARS-CoV, MERS-CoV, and SARS-CoV-2 do not seem to be that well adapted to humans, but rather their animal hosts with the occasional spillover to the human population [147]. It was also shown that MERS-CoV originated in bats, with dromedary camels being the main intermediate reservoir hosts [148,149]. Similarly, SARS-CoV and SARS-CoV-2 were shown to be of probable bat origin, with other wild animals acting as intermediate hosts, such as masked palm civets and pangolins, respectively [15,150,151]. However, these animals most likely represent only transient and accidental hosts, while the circulation of MERS in dromedary camels seems to be long-term, with cross-species transmission occurring >30 years ago [152,153].

The most likely natural reservoirs of sarbecoviruses are various horseshoe bats from the *Rhinolophidae* family, which are hosts to bat SARS-CoV [15,141,144]. However, none of the bat SARS-CoVs seem to be a direct ancestor of the SARS-CoV and SARS-CoV-2 due to relatively distant phylogenetic relationships [154,155]. It should be emphasised that both SARS-CoV and SARS-CoV-2 use angiotensin-converting enzyme II (ACE2) as a cell entry receptor [15,156]. Even though this is not a universal characteristic of the *Sarbecovirus* subgenus, several of the bat SARS-like coronaviruses, such as WIV1 and WIV16, can bind human and bat ACE2 alike [157,158]. While there is no consensus on whether the ACE2 binding ability of bat-CoV was a loss- or gain-of-function mutation, the use of this receptor seems to be a highly evolvable characteristic with single mutations in receptor binding domains of spike gene leading to severe differences in binding efficiency [140,159]. One of the closest animal viral genomes to SARS-CoV-2 was initially shown to be Bat-CoV-RaTG13, with the second closest relative being Pangolin-CoV [147,151]. The sequence identity between Bat-CoV-RaTG13 and SARS-CoV-2 is 96.2% in the overall genome and 93.1% in the S gene region [15]. On the other hand, Pangolin-CoV genome comparison showed high sequence identity with both SARS-CoV-2 (91.0%) and Bat-CoV-RaTG13 (90.6%), along even higher amino acid sequence identity with SARS-CoV-2 S protein genes (97.5%) than Bat-CoV-RaTG13 S protein genes (95.4%) [151]. Therefore, one of the putative evolutionary patterns of SARS-CoV-2 origin could be the integration of the RNA fragment of the Pangolin-CoV-2019-related strain into the Bat-CoV-RaTG13 spike protein gene [147,154]. Furthermore, even though gene transfer is rarely observed in RNA viruses, gene-by-gene horizontal transfer and recombinational analysis of SARS-CoV-2-related viruses suggest that SARS-CoV-2 could also be a close relative of the bat-CoV ZC45 and ZXC21 strains [160]. A unique feature of SARS-CoV-2, which distinguishes it from other sarbecoviruses, is the presence of a furin cleavage site between S1 and S2 subunits of spike protein beneficial for its priming and increasing the affinity for the ACE2 receptor, most likely acquired during the frequent recombination events between SARS-CoV-2 and SARS-related CoV in bat co-infection, followed by a spillover to the human population [161,162,163].

### 6.6. SARS-CoV-2 Variant Evolution

The emergence of SARS-CoV-2 in December 2019 was followed by its swift spread in the human population and continuously increasing transmissibility leading to the evolution of newly evolved variants. Mutations causing single amino acid change, such as D614G in spike protein gene and P323L in RNA-dependent RNA polymerase protein nsp12, were shown early on to be associated with higher viral loads and higher infectivity of the haplotypes that harbour them, leading to enhanced viral fitness but no change in pathogenesis [164,165]. Therefore, the evolution of SARS-CoV-2 during the first 11 months (the so-called first wave) of the pandemic was nearly neutral, requiring minimal adaptation of SARS-CoV-2 to humans [166]. However, by late 2020, a rapidly growing genomic cluster with a larger than the usual number of genetic changes was observed in the UK and labelled as the first variant of concern (VOC), Alpha (B.1.1.7), starting the second wave of the pandemic [167]. Variants of concern are defined as SARS-CoV-2 variants shown to be associated with one of the following: increase in transmissibility, increase in virulence, change in clinical disease presentation, detrimental change in epidemiology or reduced effectiveness in available therapeutics, vaccines, or diagnostics [168]. VOC Alpha was associated with multiple mutations in the spike protein gene, the most significant being N501Y within the receptor binding domain associated with the increased binding ability to ACE2 receptor, deletion H69, responsible for immune response evasion, and P681H in the furin cleavage site, which enhances spike cleavage [167,169]. Almost simultaneously, two more lineages with characteristic N501Y mutation were identified as VOC, Beta (B.1.351), and Gamma (P.1), first reported in South Africa and Brazil, respectively [170,171]. The third pandemic wave, initiated by the Delta variant (B.1.617.2), was first identified in India in late 2020 and was declared VOC in May 2021 [172]. The Delta variant seemed to lack the N501Y mutation observed in previous variants; however, multiple other mutations in spike protein resulted in a variant characterised by higher transmissibility and mortality [173,174]. The currently circulating variant of concern is Omicron (B.1.1.529), which was designated as such in November 2021 due to more than 30 mutations in the spike protein that were rarely seen in previous SARS-CoV-2 genomes. These mutations were shown to act as highly adaptive mutations and cooperatively interact, leading to an additional increase in viral infectivity and driving the fourth wave of the pandemic [175,176]. Omicron subvariants, such as BA.1, BA.2, BA.3, BA.4, and BA.5, were since identified [177,178]. It should be emphasised that all VOC originated from the initial SARS-CoV-2 genotype [179]. Visualisation of SARS-CoV-2 evolution and spread since the start of the pandemic, performed using Nextstrain, a publicly accessible bioinformatic tool for real-life tracking of pathogen evolution, shows global dominance of Omicron strains that almost completely replaced Delta and all previous variants (Figure 3) [180,181].

## 7. Conclusions and Future Perspectives

Even though evolutionary processes are generally rather slow and long-lasting when observed on a level of cellular organisms, characteristics of the RNA viruses described in this review article enable real-time monitoring of some of the key evolutionary mechanisms. The increase in genetic variability could be considered one of the prerequisites for exploring new niches, while the plasticity of RNA viruses is best described by the multitude of ways utilised for processes such as genetic recombination and mutation. The emergence of three highly pathogenic RNA viruses from the *Coronaviridae* family showed that there are still many unknowns regarding viral pathogenesis, especially events leading up to potential cross-species transmission. Furthermore, the evolution of SARS-CoV-2 and its variants demonstrated the importance of the quasispecies concept and the need to revise rather old-fashioned constructs, such as the existence of the ‘wild type’ genome, when it comes to RNA viruses. There are many unknowns regarding the further trajectory of the ongoing SARS-CoV-2 pandemic. Despite the relative success of the SARS-CoV-2 vaccines (which is out of the scope of this review), it must be emphasised that the expectations for vaccine efficiency were a lot higher during 2020 based on then-observed limited diversity between viral genomes [182]. On the other hand, a considerable boost in transmissivity and host adaptation led to a drop in infection severity distinctive for Omicron variants [183]. The highly recombinogenic nature of coronaviruses emphasises the importance of surveillance of both intermediate and reservoir host animals. Furthermore, elucidating the exact mechanisms of cross-species transfer along with deciphering the complete mutational profile of the significant VOC could be highly beneficial in predicting the trajectory of the SARS-CoV-2 pandemic and preparing for the inevitable occurrence of the next one.

## Figures and Tables

**Figure 1 viruses-15-00001-f001:**
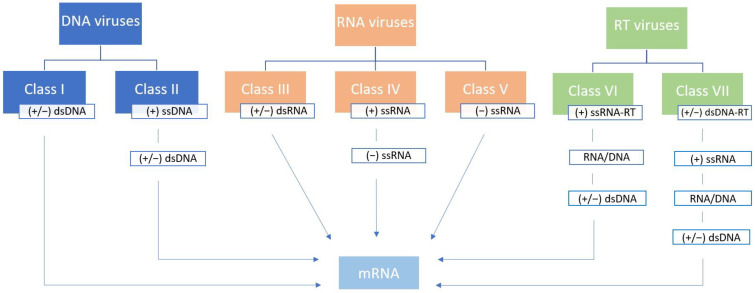
Baltimore classification of viruses. Seven classes of viruses are depicted based on the genome structure and mRNA synthesis strategy. The polarity of the genome is designated as “+” and “−” strands, or strands of both polarities (“+/−”), reverse transcribing characteristic is abbreviated as “RT”, while “ds” and “ss” indicate double-stranded and single-stranded genomes, respectively. Data are based on the original and updated Baltimore classification [4,22].

**Figure 2 viruses-15-00001-f002:**
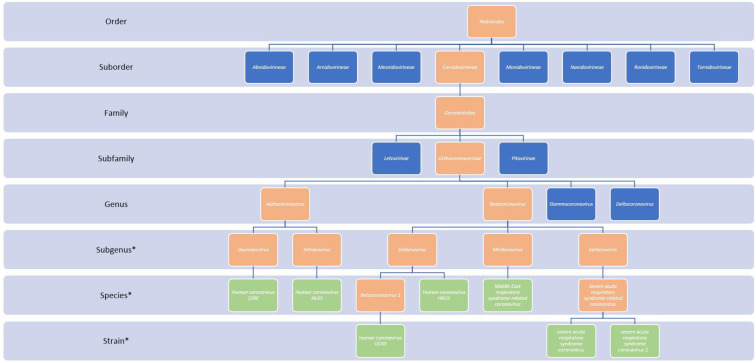
Classification of human coronaviruses. Seven currently recognised human coronavirus species are shown in green, their higher taxonomical categories are shown in orange, and other taxonomical categories are shown in blue. *: not all members of the particular taxonomical category are depicted. Data are based on the International Committee on Taxonomy of Viruses: Current Taxonomy Release 2021 and Master Species List 2021 v3 [32,109].

**Figure 3 viruses-15-00001-f003:**
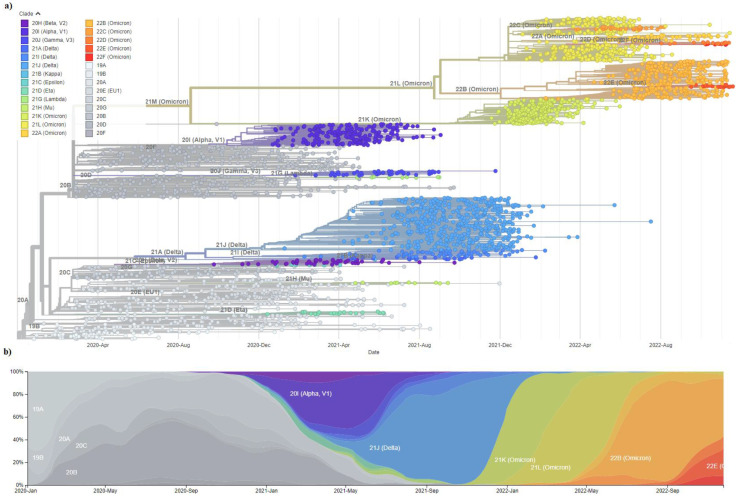
Global genomic epidemiology of severe acute respiratory syndrome coronavirus 2 (SARS-CoV-2) since the start of the coronavirus disease 2019 (COVID-19) pandemic. Visualisation of SARS-CoV-2 evolution and spread since the start of the pandemic was performed using Nextstrain (nextstrain.org), a publicly accessible bioinformatic tool for real-life tracking of pathogen evolution. A depiction of a subsample of 3059 genomes obtained between December 2019 and November 2022 from the GISAID database, with currently recognised SARS-CoV-2 variants indicated by different coloured branches on (**a**) a time-resolved phylogenetic tree. (**b**) Frequency visualisation by clade. Isolate Wuhan-Hu-1 (accession number: MN908947.3) was used as a reference. Vector images and live display can be found at: https://nextstrain.org/ncov/gisaid/global/all-time?d=tree,frequencies&p=full&showBranchLabels=all (accessed on 29 November 2022). In addition, a complete list of 3059 sequence authors was downloaded and shown in Appendix A as a tabs-separated value (TSV) file. Vector images are licensed with Attribution 4.0. International (CC BY 4.0) license [180,181].

## Data Availability

The data presented in this study are available in Appendix A.

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
