# Peer review of "A Glimpse on the Evolution of RNA Viruses: Implications and Lessons from SARS-CoV-2"

_viruses, 2022, doi:10.3390/v15010001_

Round 1

Reviewer 1 Report

RNA viruses exhibit high genetic variability by means of mutation, recombination and reassortment. This feature of RNA viruses allows rapid virus evolution, which leads to changes in viral transmissivity and difficulties in viral eradication. Given the ongoing SARS-CoV-2 pandemic, it is extremely important to fundamentally understand the molecular mechanisms of variability and evolution of viral RNA genomes. In this review, Simicic and Zidovec-Lepej extensively discussed the general ways that exploited by RNA viruses to increase genetic variability and the underlying molecular mechanisms. The authors then focused on SARS-CoV-2 virus and introduced origin of this virus and mechanisms of SARS-CoV-2 evolution. The authors also discussed challenges and consequences of SARS-CoC-2 variance on human efforts fighting this virus. Overall, this is a very informative and well-written review. I read it with lots of joys and strongly recommend for publication. I only have a few minor editorial points as listed below.

It is not clear what the dsDNA (+/-) represents in Figure 1 and it should be defined in the main text.

Line 153, it should be "3.4.", not "3.2.".

Line 512, it should be Figure 3 instead of Figure 1.

Line 537-540, this sentence is ambiguous, please rephrase for clarity.

Is there any reassortment observed in coronaviruses? If yes, it should be discussed in the manuscript.

It is difficult to read the words in Figure 2 and Figure 3, please increase the font size.

Author Response

Reviewer 1

RNA viruses exhibit high genetic variability by means of mutation, recombination and reassortment. This feature of RNA viruses allows rapid virus evolution, which leads to changes in viral transmissivity and difficulties in viral eradication. Given the ongoing SARS-CoV-2 pandemic, it is extremely important to fundamentally understand the molecular mechanisms of variability and evolution of viral RNA genomes. In this review, Simicic and Zidovec-Lepej extensively discussed the general ways that exploited by RNA viruses to increase genetic variability and the underlying molecular mechanisms. The authors then focused on SARS-CoV-2 virus and introduced origin of this virus and mechanisms of SARS-CoV-2 evolution. The authors also discussed challenges and consequences of SARS-CoC-2 variance on human efforts fighting this virus. Overall, this is a very informative and well-written review. I read it with lots of joys and strongly recommend for publication. I only have a few minor editorial points as listed below.

Thank you very much for your kind suggestions and reccomendations. Here are our responses:

  1. It is not clear what the dsDNA (+/-) represents in Figure 1 and it should be defined in the main text.

Definition of abbreviations such as “+” , “-”, “+/-”, “ds”, “ss” was included in the description of Figure 1, as well as in the main text.

  1. Line 153, it should be "3.4.", not "3.2.".

The chapter subtitle in line 153 was corrected to „3.4.”

  1. Line 512, it should be Figure 3 instead of Figure 1.

Figure number in line 512 was corrected to Figure 3 instead of Figure 1.

  1. Line 537-540, this sentence is ambiguous, please rephrase for clarity.

The sentence in lines 537-540 was rephrased and shortened for better clarity.

  1. Is there any reassortment observed in coronaviruses? If yes, it should be discussed in the manuscript.

Since coronaviruses posses no physical segmentation of genome they cannot reassort, therefore this wasn't discussed further in the manuscript.

  1. It is difficult to read the words in Figure 2 and Figure 3, please increase the font size.

For better visibility of Figure 2 and Figure 3, we suggest that they are reprinted on separate pages in landscape format, which will significantly improve readability. Therefore, the layout change on pages 9 (Figure 2) and 14 (Figure 3) was performed.     

Reviewer 2 Report

This is a nice manuscript with well-organized structure, adequate scientific proofs and enlightened opinions. Some questions and/or mistakes are listed below:

1. The current title may be a little too big and oversell. It could be narrowed down and simplified a little bit. A more appropriate title could be “A glimpse on evolution of RNA virus: implications and lessons from SARS-CoV-2” or something alike. However, this is just my personal opinion and the authors should feel free to take it or not.

2. In section 3.3, the authors talked about “RNA regulatory process”, which usually could be divided into three main parts: replication, transcription and the following up translation. However, there was no discussion about the replication process. The replication of viral RNA, which usually applying different strategy from transcription of mRNA, is indispensable, as the ultimate goal of virus is to replicate itself.

3. In line 153, the section number is mislabeled, it should be 3.4 instead of 3.2. As we are on this topic, I found this section is more closely connected with section 4 and can be re-positioned. Again, just a personal opinion.

4. In section 4, the authors mentioned “gene transfers” (line 191) but chose not to elaborate it. As gene transfers is a very interesting event occurring in the evolution of RNA and there are some discussions about gene transfers on SARS-CoV-2 (especially some non-coding parts in its genome), the authors could add some sentences in the end of this section to discuss it. This suggestion is not obligatory and the authors should feel free to make the decision.

5. In line 311 and 312, the word “related” is questionable. Usually, SARSr-CoV represents a sub group (or species) of CoVs that are evolutionarily related to SARS-CoV (as the illustrated in Figure 2), while the causative agent is just called SARS-CoV(SARS-CoV-2).

6. Some misspells and typos: line 224, “class I”; line 467, “SarS-CoV-2”; line 497, “2011”; line 508, “Nexststrain”; line 512, figure number;

7. The font size of Figure 2 could be optimized, for now it is too small to recognize.

Author Response

Reviewer 2

This is a nice manuscript with well-organized structure, adequate scientific proofs and enlightened opinions. Some questions and/or mistakes are listed below:

Thank you very much for your kind suggestions and reccomendations. Here are our responses:

  1. The current title may be a little too big and oversell. It could be narrowed down and simplified a little bit. A more appropriate title could be “A glimpse on evolution of RNA virus: implications and lessons from SARS-CoV-2” or something alike. However, this is just my personal opinion and the authors should feel free to take it or not.

We agree with everything mentioned above and therefore we changed the manuscript title to: “A glimpse on evolution of RNA viruses: implications and lessons from SARS-CoV-2”

  1. In section 3.3, the authors talked about “RNA regulatory process”, which usually could be divided into three main parts: replication, transcription and the following up translation. However, there was no discussion about the replication process. The replication of viral RNA, which usually applying different strategy from transcription of mRNA, is indispensable, as the ultimate goal of virus is to replicate itself.

We added a part on replication strategies in section 3.3. (“RNA regulatory processes”) where we emphasized how they differ from transcription processes. Since section 4.1. (“Mutation”) had a part dedicated to replication and RNA-dependent RNA polymerase, we re-positioned it to section 3.3. where it is now better suited.  

  1. In line 153, the section number is mislabeled, it should be 3.4 instead of 3.2. As we are on this topic, I found this section is more closely connected with section 4 and can be re-positioned. Again, just a personal opinion.

The chapter subtitle in line 153 was corrected to „3.4.”. We find that this chapter (“Quasispecies concept”) is perhaps better suited to the section 3 (“RNA virus characteristics”) since the quasispecies nature of RNA viruses is one of their crucial characteristics and the consequence, rather than cause, of the mechanisms of RNA virus variation which are then explored in the following chapters of section 4.

  1. In section 4, the authors mentioned “gene transfers” (line 191) but chose not to elaborate it. As gene transfers is a very interesting event occurring in the evolution of RNA and there are some discussions about gene transfers on SARS-CoV-2 (especially some non-coding parts in its genome), the authors could add some sentences in the end of this section to discuss it. This suggestion is not obligatory and the authors should feel free to make the decision.

We did not include a separate chapter on gene transfers due to its relatively rare occurrence in RNA viruses and focused instead primarily on the most common mechanisms of RNA virus variation. However, we did mention new discoveries on gene transfer in section 6.5. as a part of the narrative on the origin of SARS-CoV-2. 

  1. In line 311 and 312, the word “related” is questionable. Usually, SARSr-CoV represents a sub group (or species) of CoVs that are evolutionarily related to SARS-CoV (as the illustrated in Figure 2), while the causative agent is just called SARS-CoV(SARS-CoV-2).

We corrected the full names of SARS-CoV and SARS-CoV-2 in the manuscript text to match the correct terms indicated in Figure 2.

  1. Some misspells and typos: line 224, “class I”; line 467, “SarS-CoV-2”; line 497, “2011”; line 508, “Nexststrain”; line 512, figure number;

All of the above mentioned misspells and typos were corrected.

  1. The font size of Figure 2 could be optimized, for now it is too small to recognize.

For better visibility of Figure 2 (and Figure 3) we suggest that they are reprinted on separate pages in landscape format which will greatly improve readability. Therefore, the layout change on pages 9 (Figure 2) and 14 (Figure 3) was performed.